# Acoustic Individual Identification in Birds Based on the Band-Limited Phase-Only Correlation Function

**Angel David Pedroza** [1,†], **José I. De la Rosa** [1,*,†], **Rogelio Rosas** [2,†], **Aldonso Becerra** [1], **Jesús Villa** [1], **Gamaliel Moreno** [1], **Efrén González** [1] **and Daniel Alaniz** [1]

1  Unidad Académica de Ingeniería Eléctrica, Universidad Autónoma de Zacatecas, Avenida Ramón López Velarde 801, Zacatecas 98000, Mexico; p.a.d_16@hotmail.com (A.D.P.); a7donso@uaz.edu.mx (A.B.); jvillah@uaz.edu.mx (J.V.); gamalielmch@gmail.com (G.M.); gonzalez_efren@hotmail.com (E.G.) dalaniz@uaz.edu.mx (D.A.)
2  Unidad Académica de Ciencias Biológicas, Universidad Autónoma de Zacatecas, Carretera a la Bufa, Zacatecas 98000, Mexico; rogrosas@uaz.edu.mx
*  Correspondence: vargasj@uaz.edu.mx
†  These authors contributed equally to this work.

**Abstract:** A new technique based on the Band-Limited Phase-Only Correlation (BLPOC) function to deal with acoustic individual identification is proposed in this paper. This is a biometric technique suitable for limited data individual bird identification. The main advantage of this new technique, in contrast to traditional algorithms where the use of large-scale datasets is assumed, is its ability to identify individuals by the use of only two samples from the bird species. The proposed technique has two variants (depending on the method used to analyze and extract the bird vocalization from records): automatic individual verification algorithm and semi-automatic individual verification algorithm. The evaluation of the automatic algorithm shows an average precision that is over 80% for the identification comparatives. It is shown that the efficiencies of the algorithms depend on the complexity of the vocalizations.

**Keywords:** bioacoustics; speech recognition; bird identification; individuals classification; limited data

## 1. Introduction

Biodiversity comprises several levels of biological variation, from genes to ecosystems, where species richness and abundance are two of its important features [1,2]. The density of birds in an area is an indicator that can be used to determine bird survival, reproduction, dispersion, and population growth [3]. This parameter is measured through a search methodology traditionally based on point counts, linear transects, and mist nets techniques [4–6]. The point counts and linear transects techniques are used to monitor population change by identifying birds by sight and/or their characteristic sound. In this sense, for the reliability of the evaluation, some conditions must be guaranteed such as the lack of movement of birds before detection and the statistical independence of the counts [7]. Mist nets are used to capture individuals and determine features such as sex, age, and bird survival [8]. Commonly, artificial marks are used on the captured birds to determine, among others, the dispersion of their habitats and the mapping of their territories [9]. Although these techniques have shown good results, the use of non-standard methods and the limited availability of bird experts reduce their effectiveness [5].

Bioacoustics is an alternative method of non-invasive monitoring that studies biodiversity based on the relationship between animal species and their sounds [10–12]. The study of these signals

is important in diverse biology areas, such as behavior, physiology, and evolution, among others [13,14]. Specifically, the acoustic individual identification in birds involves the extraction of features from the vocalizations (song or call) of a specific individual that allows its identification among other vocalizations from other same-species individuals (or even from other bird species) [15]. Some of the statistical techniques commonly used for speaker recognition can also be applied to the individual identification of animals [16,17]. Although these techniques have good performance on automatic speaker recognition through the use of large-scale datasets, some of them have an inadequate performance in limited data conditions (small datasets) [18]. In this sense, for the acoustic identification of bird individuals in real-field conditions, the amount (and quality) of vocalizations from an individual makes it difficult to apply any statistical methods efficiently. It is nowadays a complex challenge.

This paper proposes a new technique to deal with the acoustic individual identification in birds based on the Band-Limited Phase-Only Correlation (BLPOC) function [19]. The BLPOC function is a high accuracy biometric technique traditionally implemented for human identification by fingerprints recently applied in speaker verification [20]. In this work, the (BLPOC) function is adapted to process bird vocalizations as a limited data individual identification method. This is a proof-of-concept whose main advantage, in contrast to traditional algorithms where the use of large-scale datasets is assumed, is its ability to provide an individual identification result by only verifying two samples from the bird species. The proposed method has two variants (depending on the method used to extract bird vocalization from records): one that is automatic and another that is considered as semi-automatic.

This paper is organized as follows, Section 2 explains the basic theory of the BLPOC function. Section 3 details the proposed acoustic individual identification algorithm. Section 4 shows the results of the algorithm evaluation. Finally, Section 5 discusses the findings and Section 6 provides some conclusions.

## 2. Band-Limited Phase-Only Correlation (BLPOC)

The Phase-Only Correlation (POC) function is an efficient technique commonly applied on image matching tasks as a measure of similarity between two sequences. However, when the main information of a given signal is clustered in a frequency band, the meaningless phase components of the signal reduce the efficiency of the method. In this sense, the Band-Limited Phase-Only Correlation (BLPOC) eliminates meaningless information by extracting an effective region obtained from the compared sequences [21–26]. The basic theory of the BLPOC function is described below.

Given two $N$-point 1D sequences ($f(n)$ and $g(n)$), for which the index ranges are $n = 0, \ldots, N-1$, the 1D Discrete Fourier Transform (1D DFT) of each sequence ($F(k)$ and $G(k)$, respectively) is calculated as [27]:

$$F(k) = \sum_{n=0}^{N-1} f(n) W_N^{kn} = A_F(k) \exp(j\theta_F(k)), \tag{1}$$

$$G(k) = \sum_{n=0}^{N-1} g(n) W_N^{kn} = A_G(k) \exp(j\theta_G(k)), \tag{2}$$

where $k = 0, \ldots, N-1$, and $W_N = \exp(-j2\pi/N)$; amplitude components are denoted as $A_F(k)$ and $A_G(k)$; and $\theta_F(k)$ and $\theta_G(k)$ denote the phase components. Next, the cross-phase spectrum ($Rn_{FG}(k)$) between $F(k)$ and $G(k)$ is defined as:

$$Rn_{FG}(k) = \frac{F(k)\overline{G(k)}}{\|F(k)\overline{G(k)}\|}, \tag{3}$$

where $\overline{G(k)}$ denotes the complex conjugate of $G(k)$. An effective region is calculated to eliminate the meaningless information in the cross-phase spectrum (depending on the input sequence) [19,26]. Thus,

assuming the zero frequency component of the input sequence $F(k)$ is shifted at the center of the spectrum, the range of the effective region (defined by the limits $F_1$ and $F_2$) is:

$$k = F_1, \ldots, F_2, \tag{4}$$

where $0 \leq F_1 \leq F_2$ and $F_1 \leq F_2 \leq N - 1$. Finally, the BLPOC function ($\hat{r}_{fg}(n)$) is the 1D Inverse Discrete Fourier Transform (1D IDFT) of $Rn_{FG}(k)$, which is calculated by:

$$\hat{r}_{fg}(n) = \frac{1}{L} \sum_{k=F_1}^{F_2} Rn_{FG}(k) W_L^{-kn}, \tag{5}$$

where $L$ is the effective size of the region defined as $L = (F_2 - F_1) + 1$. The BLPOC function is a measure of similarity between sequences and, in this sense, if $f(n)$ and $g(n)$ are the same sequence, the resulting BLPOC function is the Kronecker's delta function ($\delta(n)$); on the other hand, if $f(n)$ and $g(n)$ are not the same sequence, the resulting similarity is computed through a matching score.

## 3. Acoustic Individual Identification in Birds

A vocalization database of a few individuals is necessary to evaluate the proposed methodology. For this purpose, several audio files were collected from two free online databases:

- The bird sound library of Mexico (from INECOL) [28]: A small database used for ornithology research and the diffusion of audio records of bird species in Mexico.
- Xeno-canto [29]: A citizen-science database with about 500k audio records of vocalization of bird species from around the world.

Although databases have different audio records, some of them do not offer specific features (e.g., the type of vocalization or information about if the records are from more than one individual). The acoustic information shown in Table 1 was obtained from the databases for different species in order to use it in the proposed algorithm (all birds in the database are Passeriformes). There is not a special consideration about the selection of bird species. Given that there is a wide variety of songs available in databases, a quick search was made (audios that were easy to identify acoustically) to test the system.

**Table 1.** List of bird species and records in the database.

| Family | Species | Number of Individuals | Number of Vocalizations from Each Individual |
|--------|---------|----------------------|---------------------------------------------|
| Furnariidae | (**a**) *Clibanornis rubiginosus* (P.L.Sclater, 1857) | 4 | 4,4,27,6 |
| | (**b**) *Synallaxis erythrothorax* (P.L.Sclater, 1855) | 6 | 7,15,7,5,27,9 |
| Cardinalidae | (**c**) *Cardinalis cardinalis* (Linnaeus, 1758) | 3 | 17,2,22 |
| Thamnophilidae | (**d**) *Cercomacra tyrannina* (P.L.Sclater, 1855) | 4 | 7,4,2,3 |
| Troglodytidae | (**e**) *Hylorchilus sumichrasti* (Lawrence, 1871) | 5 | 4,6,4,4,2 |
| Tyrannidae | (**f**) *Myiozetetes similis* (Spix, 1825) | 6 | 10,28,8,13,7,8 |
| | (**g**) *Pitangus sulphuratus* (Linnaeus, 1766) | 3 | 5,5,6 |
| Icteridae | (**h**) *Psarocolius montezuma* (Lesson, 1830) | 5 | 5,3,2,2,2 |
| Psittacidae | (**i**) *Amazona viridigenalis* (Cassin, 1853) | 2 | 6,3 |

Table 1 gives information about the number of vocalizations extracted from original audio files. The variability of vocalizations correspond to the variability of audio files available from a specific individual and the number of specific vocalizations in each audio file.

The proposed method uses a pre-stored audio record of a bird vocalizations (call or song) to identify whether an input audio record belongs to the same individual (genuine matching) or not

(impostor matching). As mentioned in the Introduction, the proposed technique has two variants (depending on the method used to analyze and extract the bird vocalization from records): automatic individual verification algorithm and semi-automatic individual verification algorithm. The process is depicted in Figure 1.

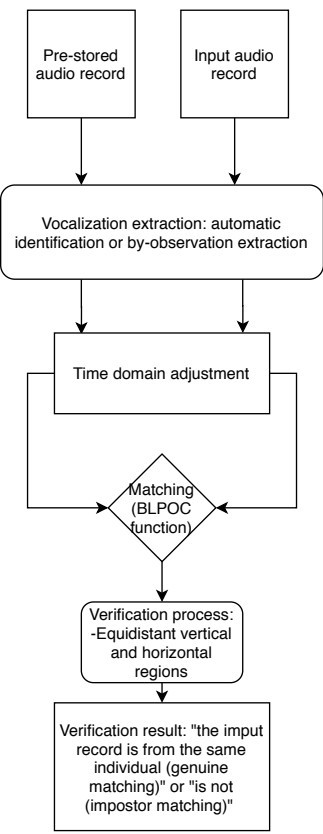

**Figure 1.** Proposed acoustic individual identification algorithm. Each arrow in the diagram represents a signal processed.

### 3.1. Automatic Individual Verification Algorithm

#### 3.1.1. Automatic Identification of Bird Vocalization

It is very common the audio records of bird vocalizations have periods of silence and other background sounds. For this reason, the segments of the signal corresponding to bird vocalization need to be extracted (see Figure 2). Let $f(n)$ be an audio record of a bird vocalization. Based on a voice activity detection (VAD) algorithm [30], the bird vocalization region ($SF$) is extracted as follows: (i) Compute the energy average of the whole signal using $E = \frac{1}{N} \sum_{n=0}^{N-1} |f(n)|^2$. (ii) Divide $f(n)$ into $M$ frames ($fr_j(i)$) of $m$ equal samples according to sampling frequency. (iii) Compute the energy average of each $j$th frame as $e(j) = \frac{1}{m} \sum_{i=1}^{m} |fr_j(i)|^2$. (iv) Determine a suitable whole energy threshold value ($thr_E$) depending on the signal to noise ratio (SNR). (v) Identify the bird vocalization frames by using Equations (6) and (7):

$$SF_s = first(\{fr_j(i)|e(j) \geq thr_E, 1 \leq j \leq M, 1 \leq i \leq m\}), \tag{6}$$

$$SF_f = last(\{fr_j(i))|e(j) \geq thr_E, 1 \leq j \leq M, 1 \leq i \leq m\}), \tag{7}$$

where $SF_s$ and $SF_f$ are the first and last detected bird vocalization frames in $f(n)$. (vi) Define the bird vocalization region according to:

$$SF = [SF_s, \ldots, SF_f]. \tag{8}$$

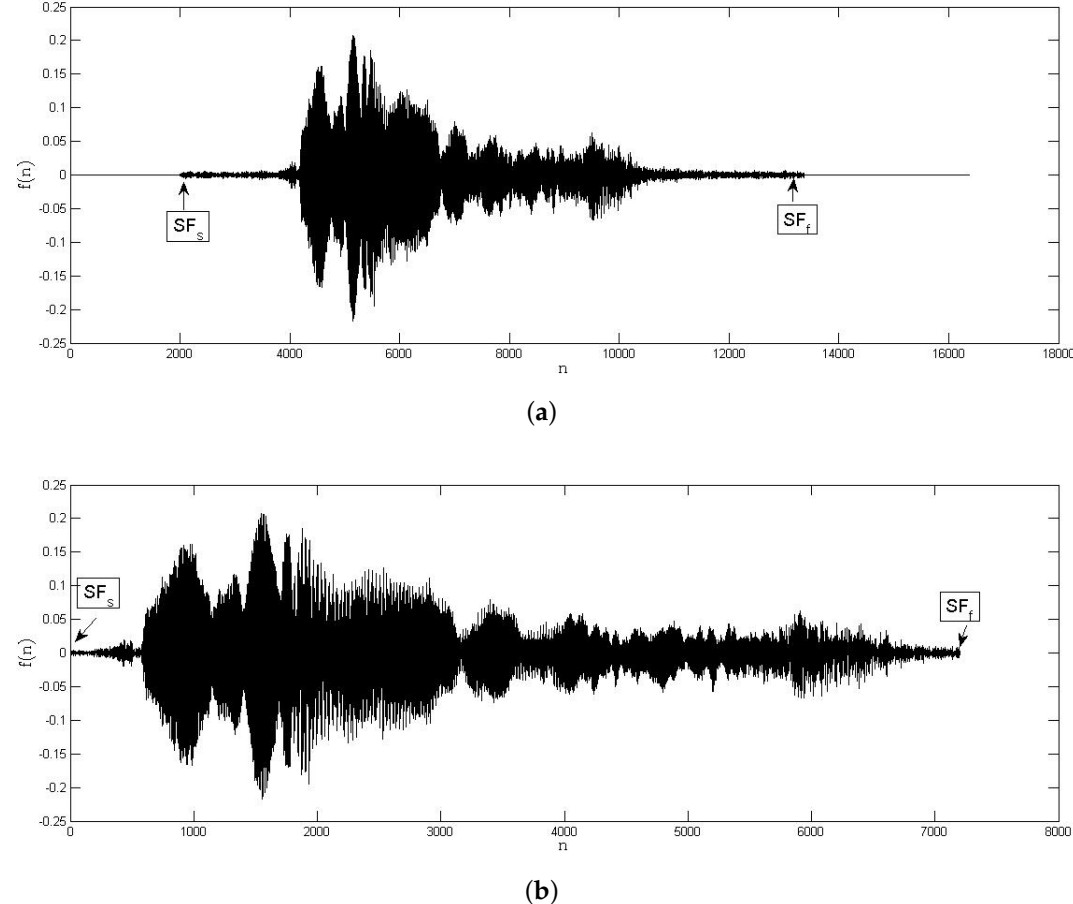

**Figure 2.** Bird vocalization automatic identification: audio record of a bird vocalization (**a**) before and (**b**) after automatic region extraction.

The whole energy threshold value used in the experiments was $thr_E = 0.001 * E$.

### 3.1.2. Time Domain Adjustment and Matching

Before the matching between the records of the bird vocalization is carried out, an adjustment to reduce the time domain variability in the sequences is applied (see Figure 3 a, b, before and after time domain adjustment, respectively). This process is as follows: (i) Compare the number of samples between the pre-stored and the input record. (ii) Adjust the time domain variability by a zero padding at the end of the shortest record (according to the number of samples in each signal).

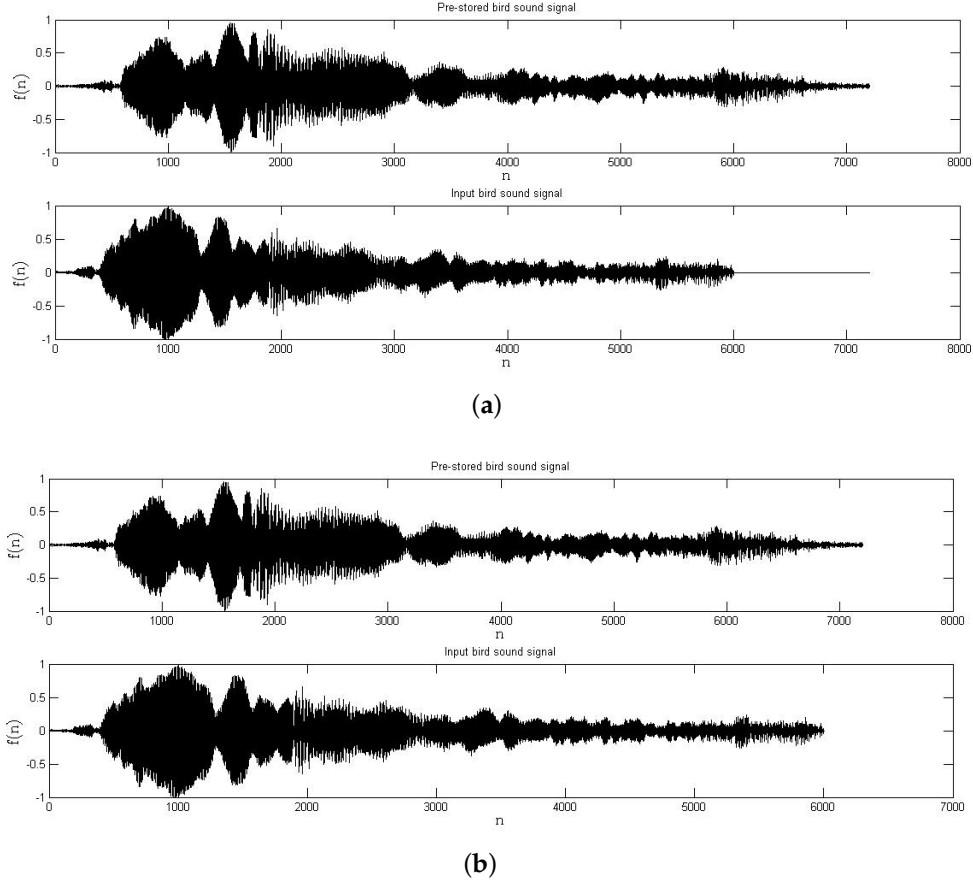

**Figure 3.** Example of an audio record of a bird vocalization (**a**) before and (**b**) after time domain adjustment.

The inherent frequency band is extracted traditionally by analyzing the DFT of the pre-stored sequence [19]. To avoid inconsistencies in the BLPOC function, the proposed algorithm detects the inherent frequency band in the cross-phase spectrum of the signals. Thus, when the compared signals have the same length, the effective frequency band can be obtained as follows (see Figure 4): (i) By using Equations (1)–(3), calculate the cross-phase spectrum ($Rn_{FG}(k)$) between bird vocalizations. (ii) Compute the Normal Power Spectral Density ($NPSD$) by using the following equation:

$$NPSD = \frac{Rn_{FG}\overline{Rn_{FG}}}{\max(Rn_{FG}\overline{Rn_{FG}})},$$ (9)

where $\overline{Rn_{FG}}$ denotes the complex conjugated of $Rn_{FG}$. (iii) Compute and store the frequency band index by the equations:

$$F_1 = \min(\{i|NPSD(i) \geq thr_p, 0 \leq i \leq N-1\}),$$ (10)

$$F_2 = \max(\{i|NPSD(i) \geq thr_p, 0 \leq i \leq N-1\}),$$ (11)

where $thr_p$ is a suitable NPSD threshold. The threshold value used in the experiments was $thr_p = 0.001$. Finally, the matching between signals is calculated using Equation (5) and the zero frequency component of $\hat{r}_{fg}$ sequence is shifted at the center of the spectrum.

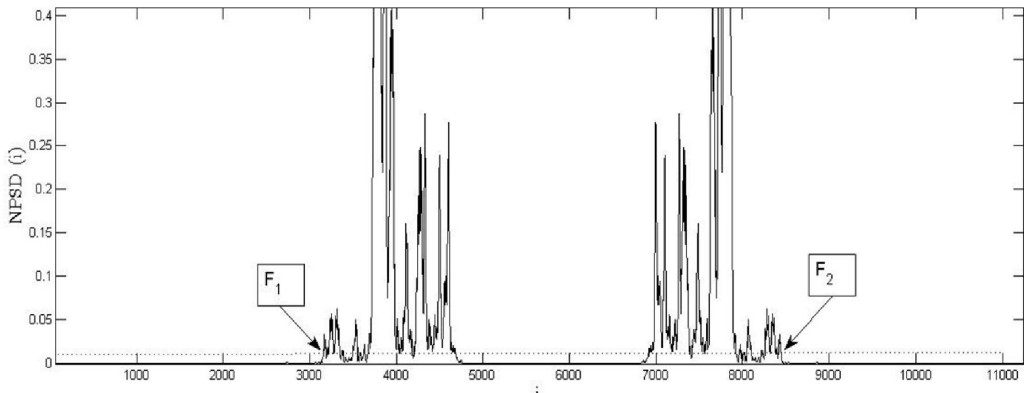

**Figure 4.** Example of the effective frequency band of the cross-phase spectrum of the compared signals. Dashed line denotes the proposed threshold ($thr_p$).

### 3.2. Semi-Automatic Individual Verification Algorithm

#### 3.2.1. By-Observation Vocalization Extraction

Usually, the skills of the observers are affected by the conditions of the observations (especially when vegetation height increase or when bare ground decreases) [4,5]. Since, in some cases, the vocalization of an individual could be the unique source for identification, the quality of the record plays an important role. The sound variability between bird species makes it difficult to have an automatic extraction algorithm of the bird vocalization from records. In this context, it is proposed to extract the main acoustic information from the audio records through a visual filter. This process is based on user (observer) selection of two spectrogram points of an audio record to generate a region of interest where the most important time–frequency domain information is located (as one can see in Figure 5).

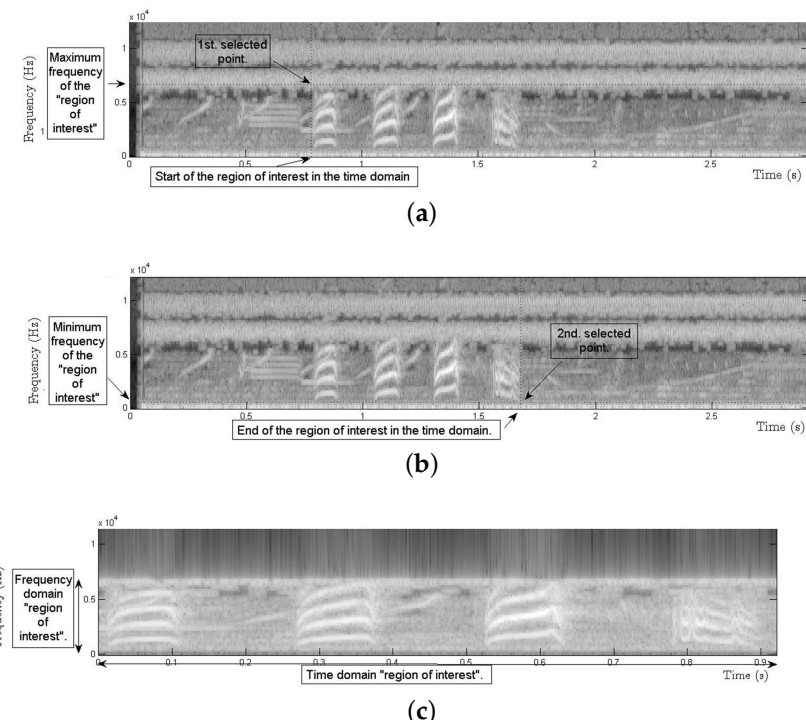

**Figure 5.** Example of by-observation vocalization extraction: (**a**) first selected point; (**b**) second selected point; and (**c**) region extracted by the user selection.

Consider $f(n)$ as an audio record from a bird vocalization. To extract the most important information from the record, the following tasks were carried out: (i) Compute the 1D DFT of the sequence using Equation (1). (ii) From the region of interest selected in the spectrogram, convert the selected points of the frequency domain to coordinates in $F(k)$. (iii) Calculate the minimum in the sequence $F(k)$ and replace the magnitude value in the frequencies of $F(k)$ outside the region of interest limited by the selected points (see Figure 6). (iv) Calculate its 1D IDFT according to:

$$f''(n) = \frac{1}{N} \sum_{k=0}^{N-1} F(k) W_N^{-kn},$$ (12)

where $f''(n)$ is the audio record in the frequency region of interest of the bird vocalization. (v) From the selected region of interest in the spectrogram, convert the selected points of the time domain to coordinates of the sequence $f''(n)$. (vi) After a normalization, generate an audio playback of the signal $(f'(n))$ (as can be seen in Figure 7). Repeat the process until the resulting audio record has only the desired information.

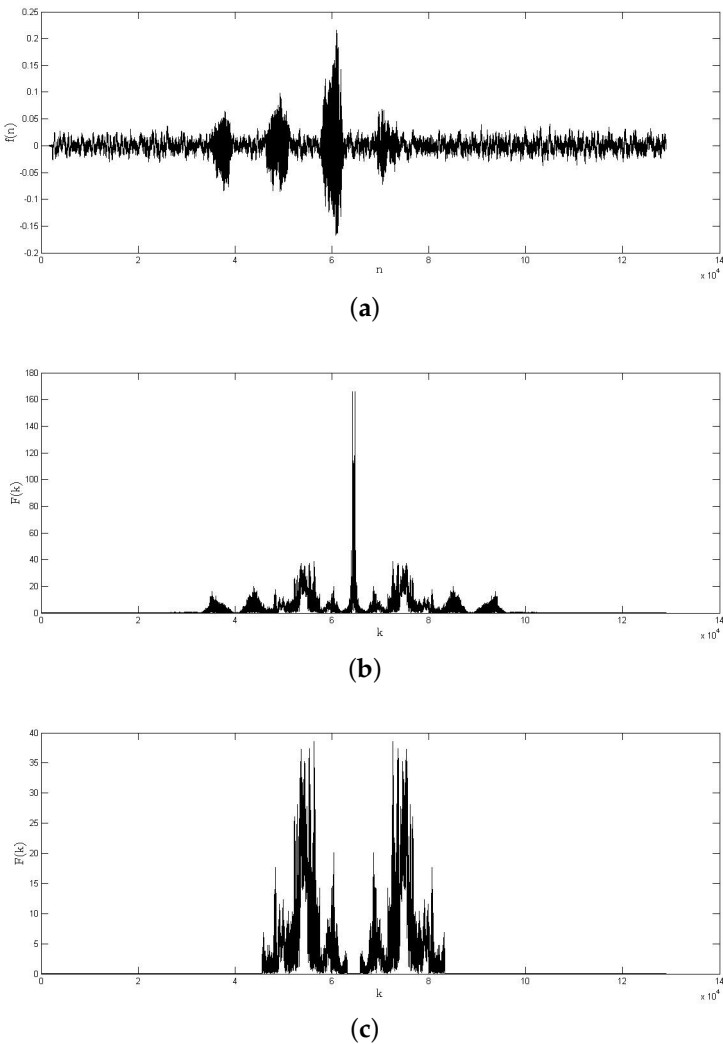

(a)

(b)

(c)

**Figure 6.** Example of a frequency region of interest extraction: (**a**) audio record from a bird vocalization; (**b**) 1D DFT of the bird vocalization; and (**c**) frequency region of interest extracted.

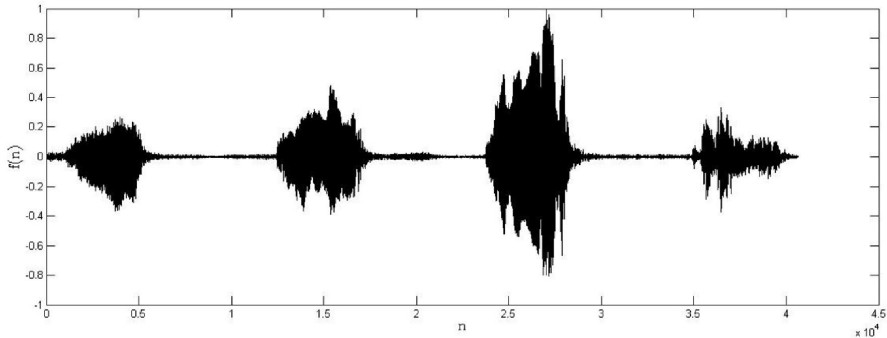

**Figure 7.** Audio record from a bird vocalization after by-observation region extraction.

### 3.2.2. Time Domain Adjustment and Matching

The time domain adjustment and matching steps are the same as those described in Section 3.1.2.

### 3.3. Individual Verification Decision

The height of the max peak in the BLPOC function can be used as a measure of similarity in human identification by image matching; however, for the individual verification through bird vocalizations, some other discrimination features (to determine a genuine and impostor matching) need to be calculated. Since the BLPOC function has multiple shapes, depending on the verified vocalizations (see Figure 8), the proposed criterion for evaluating the BLPOC function is an automatic parametric verification process based on the BLPOC function shape from genuine matching.

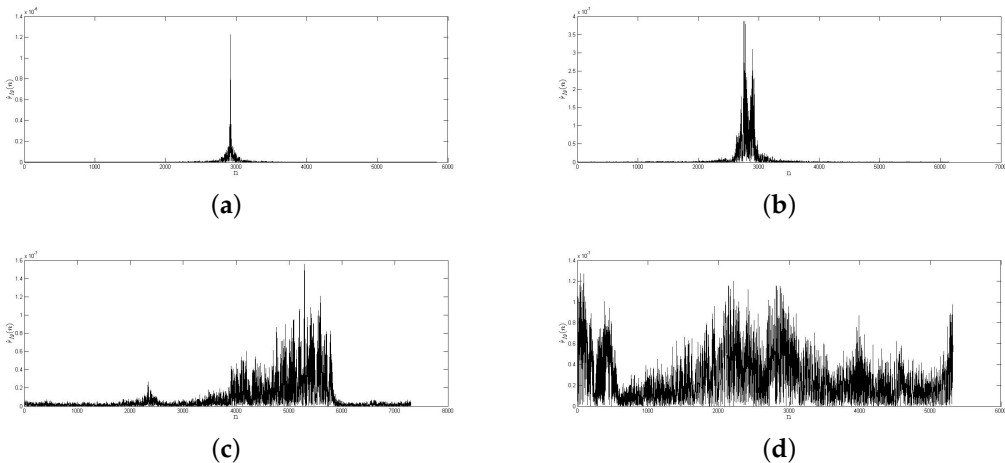

**Figure 8.** Examples of genuine and impostor matching: (**a**) BLPOC function between two identical bird vocalizations from the same individual; (**b**) BLPOC function between two similar vocalizations from the same individual; (**c**) BLPOC function between two similar vocalizations from two different individuals from same bird species; and (**d**) BLPOC function between two different individuals from different bird species.

### 3.3.1. Verification Process through Equidistant Vertical Regions

The first step in the verification process is the evaluation of the location of the max peak in the BLPOC function by the following process (see also Figure 9): (i) determine the number of samples of the BLPOC function. (ii) Calculate $m + 1$ equidistant indices ($ei(i)$) between samples. (iii) Calculate $i$th equidistant vertical regions ($evr_i$) according to:

$$evr_i = [\hat{r}_{fg}(ei(i)), \ldots, \hat{r}_{fg}(ei(i+1))]. \tag{13}$$

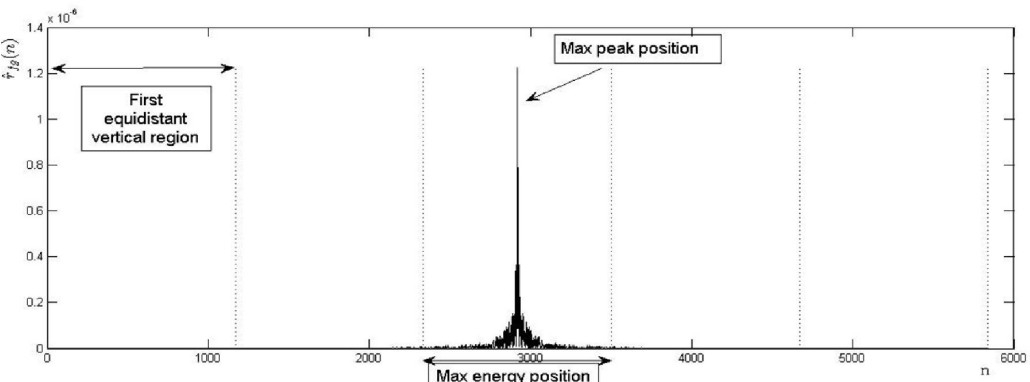

**Figure 9.** Accepted shape of the BLPOC function through equidistant vertical regions.

(iv) Calculate the index of the BLPOC function max peak ($P_M$) and determine the equidistant vertical region to which it belongs. (v) Compare the identified region with the acceptance equidistant vertical region. The number of equidistant indices in the experiments was a constant value $m = 5$ and, on the other hand, the acceptance region was the third equidistant vertical region.

However, since the BLPOC function can take many shapes, the next step is to set an acceptance shape in the equidistant vertical regions. In this sense, for genuine matching, energy is concentrated in the region of the max peak of the BLPOC function. To identify this shape in the BLPOC function, the next steps can be followed (see Figure 9): (i) Calculate the energy of the BLPOC function ($E_i$) in the $i$th equidistant vertical region ($evr_i$) according to:

$$E_i = \frac{1}{k} \sum_{j=1}^{k} |evr_i(j)|^2,$$

(14)

where $i = 1, 2, \ldots, m$ and $k$ is the number of samples in the $evr_i$. (ii), Identify the equidistant vertical region with the max energy. (iii) Compare the identified region with the acceptance location of the max energy. In the experiments, the location of acceptance of the maximum energy in the BLPOC function was the third equidistant vertical region (acceptance location of the maximum peak in the BLPOC function). Examples of BLPOC functions verification process through equidistant vertical regions are shown in Figure 10.

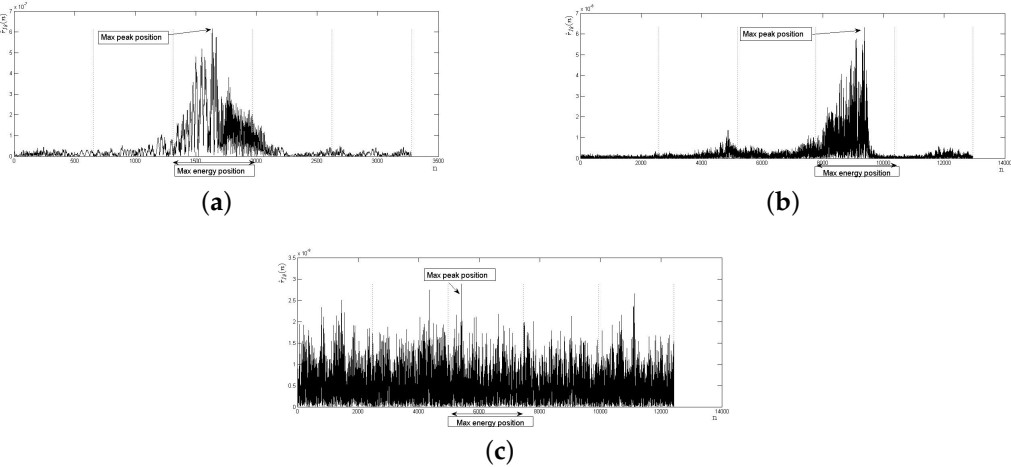

**Figure 10.** Examples of the BLPOC function verification process through equidistant vertical regions: (**a**) genuine matching correctly verified as accepted; (**b**) impostor matching correctly verified as rejected; and (**c**) impostor matching incorrectly verified as accepted.

### 3.3.2. Verification Process through Equidistant Horizontal Regions

To avoid verification errors (see Figure 10c), an evaluation of the magnitude of the less significant peaks in the non-acceptance vertical regions is made (see Figure 11). Evaluation is as follows: (i) Calculate the maximum and minimum peaks of the BLPOC function. (ii) Calculate $m + 1$ intervals of magnitude ($im(i)$) between the magnitude of the calculated peaks. (iii) Calculate the maximum peak at the $i$th equidistant vertical region $evr_i$ by:

$$peak(i) = \max |segv_i|. \tag{15}$$

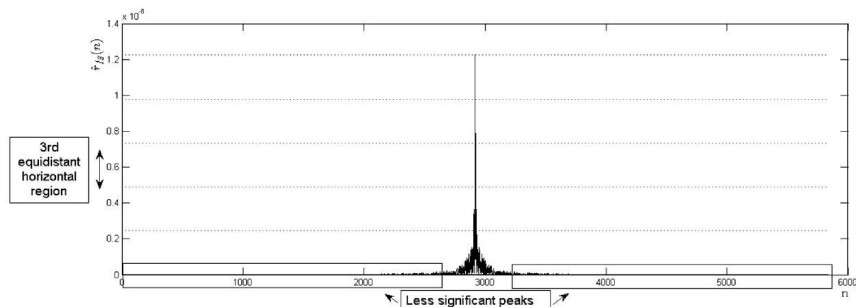

**Figure 11.** Accepted shape of the BLPOC function through equidistant horizontal regions.

(iv) Compare the identified peaks ($peak(i)$) with the maximum acceptance horizontal region. The number of intervals of magnitude in the experiments was a constant value $n = 5$. On the other hand, the maximum acceptance region was the third equidistant horizontal region, as shown in Figure 12.

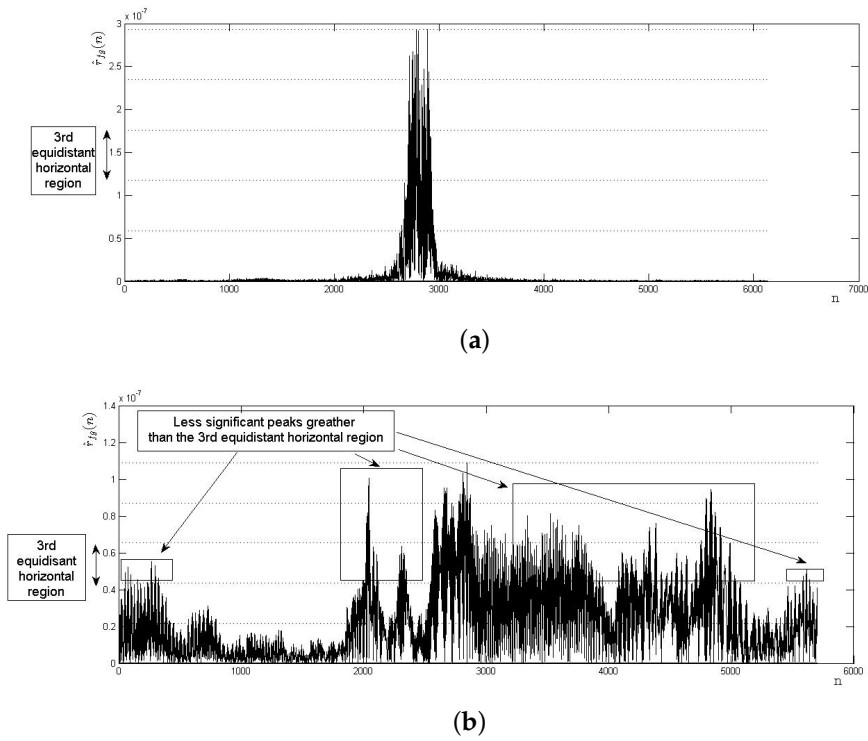

**Figure 12.** Examples of the BLPOC function verification process through equidistant horizontal regions: (**a**) a genuine matching correctly verified as accepted; and (**b**) impostor matching correctly verified as rejected.

Sub-domain deformation can introduce possible errors (see Figure 13). In the experiments, a special criterion based on an evaluation of the number of less significant maximum peaks greater than the maximum acceptance horizontal region was implemented.

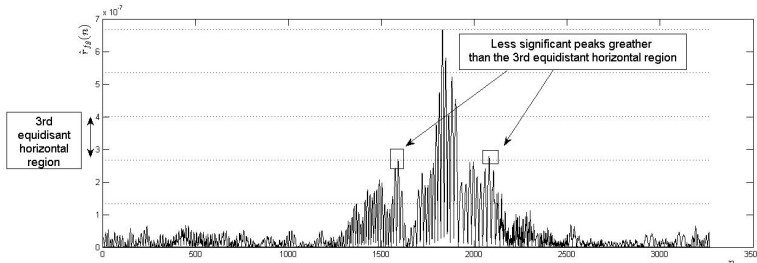

**Figure 13.** Example of possible errors due to sub-domain deformations: genuine matching incorrectly verified as rejected.

## 4. Results

Table 1 shows the list of bird species in the database [31] and the number of records. On the other hand, a subcategory of species based on the complexity of the vocalizations is proposed (Figure 14 shows two examples: (a) noncomplex vocalization; and (b) complex vocalization). As a consequence, in Table 1, the following species in the database were considered to have a complex vocalization:

- *Cercomacra tyrannina* (P.L.Sclater, 1855);
- *Hylorchilus sumichrasti* (Lawrence, 1871);
- *Pitangus sulphuratus* (Linnaeus, 1766);
- *Psarocolius montezuma* (Lesson, 1830); and
- *Amazona viridigenalis* (Cassin, 1853).

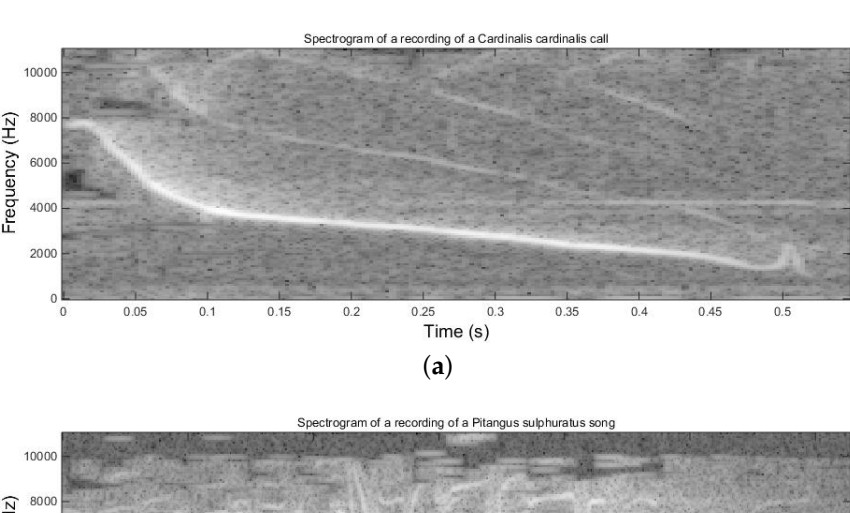

(a)

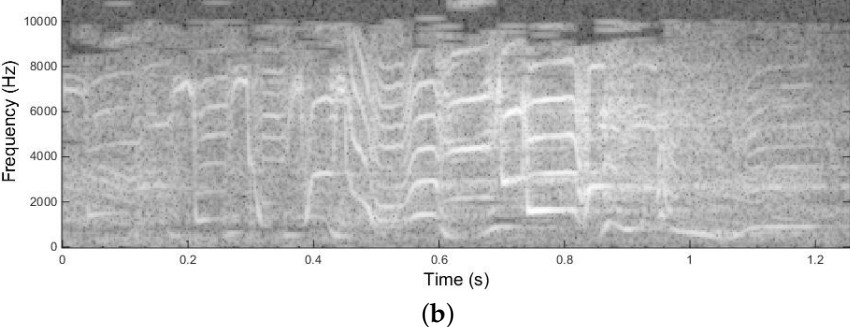

(b)

**Figure 14.** Examples of the complexity in the vocalizations: (**a**) noncomplex vocalization; and (**b**) complex vocalization.

The complexity of vocalizations is measured according to the time and frequency variations in the sound. This classification does not exist in the literature but this term is introduced in this article to make a fairer comparison for the algorithm. It is supported that it is not possible to obtain a general classification of the methodology for all species that are being recognized.

To guarantee the correct identification of individuals among other bird species individuals, an evaluation of the algorithm's discarding performance was obtained. This was made by computing the probability that a vocalization from an individual of a certain bird species is correctly rejected by the algorithm at being identified as "not generated from a specific bird species". To measure this probability, the True Rejection Rate (TRR) was calculated (the probability that an impostor individual is correctly verified as rejected). This was a combinational process where each of the individuals from a certain species was matched (using the proposed algorithm) against other individuals from other species in the database. The process was repeated for each species in the database. Some obtained results are shown in Table 2.

**Table 2.** Results of the automatic individual verification algorithm's discarding evaluation performance (%).

| VS | (a) | (b) | (c) | (d) | (e) | (f) | (g) | (h) | (i) |
|----|-----|-----|-----|-----|-----|-----|-----|-----|-----|
| **(a)** | . | 92.5 | 96.5 | 100 | 100 | 99.22 | 100 | 97.90 | 100 |
| **(b)** | 92.54 | . | 98.21 | 98.36 | 91.17 | 98.75 | 98.71 | 96.02 | 99.15 |
| **(c)** | 86.4 | 96.08 | . | 100 | 100 | 69.68 | 94.66 | 100 | 93.6 |
| **(d)** | 100 | 96.18 | 97.22 | . | 100 | 98.6 | 97.14 | 100 | 100 |
| **(e)** | 100 | 91.38 | 100 | 95.5 | . | 98.97 | 92.85 | 100 | 100 |
| **(f)** | 99.69 | 99.34 | 69.09 | 99.39 | 98.93 | . | 93.63 | 100 | 100 |
| **(g)** | 100 | 90 | 84.16 | 100 | 100 | 99.66 | . | 100 | 100 |
| **(h)** | 98.86 | 94.4 | 100 | 100 | 98.75 | 100 | 98.21 | . | 96.42 |
| **(i)** | 100 | 100 | 98.57 | 97.4 | 100 | 100 | 100 | 100 | . |

In this sense, Table 2 shows that the performance of the proposed automatic individual verification algorithm is over 90% in almost all the identification comparatives.

A second evaluation criterion was carried out through a verification test of individuals of the same bird species. To achieve this task, the following set of parameters was used:

- True acceptance (TA) is when the system verifies a genuine matching correctly.
- False acceptance (FA) is when the system verifies a genuine matching incorrectly.
- True rejection (TR) is when the system verifies an impostor matching correctly.
- False rejection (FR) is when the system verifies an impostor matching incorrectly.

This was a combinational process where each of the individuals (records) from a certain species was matched (using the proposed algorithm) against its vocalizations (genuine matching) and vocalizations from other individuals (impostor matching). The process was repeated for each species in the database. Some obtained results are shown in Tables 3 and 4.

**Table 3.** Identification evaluation of the semi-automatic algorithm.

| Species | Number of Comparatives | Parameter Results | | | |
|---|---|---|---|---|---|
| | | TA | FA | TR | FR |
| (a) | 229 | 81 | 20 | 115 | 13 |
| (b) | 300 | 102 | 20 | 143 | 35 |
| (c) | 229 | 118 | 0 | 104 | 7 |
| (d) | 28 | 4 | 18 | 3 | 3 |
| (e) | 20 | 7 | 7 | 4 | 2 |
| (f) | 300 | 19 | 13 | 227 | 41 |
| (g) | 12 | 1 | 9 | 2 | 0 |
| (h) | 21 | 7 | 8 | 3 | 3 |
| (i) | 17 | 1 | 10 | 4 | 2 |

**Table 4.** Identification evaluation of the automatic algorithm.

| Species | Number of Comparatives | Parameter Results | | | |
|---|---|---|---|---|---|
| | | TA | FA | TR | FR |
| (a) | 1681 | 511 | 286 | 764 | 120 |
| (b) | 6084 | 233 | 925 | 4391 | 535 |
| (c) | 1681 | 687 | 90 | 876 | 28 |
| (d) | 256 | 34 | 44 | 178 | 0 |
| (e) | 400 | 72 | 16 | 154 | 158 |
| (f) | 5476 | 910 | 320 | 3630 | 616 |
| (g) | 256 | 26 | 60 | 168 | 2 |
| (h) | 196 | 18 | 28 | 150 | 0 |
| (i) | 81 | 9 | 36 | 36 | 0 |

The next step was to calculate the probability that an individual would be verified as accepted (genuine matching) or rejected (impostor marching) by each of the two variants of the proposed algorithm. To measure this probability, the precision was calculated as follows (results are shown in Table 5):

$$Precision = \left( \frac{TA + TR}{TA + FA + TR + FR} \right) * 100. \tag{16}$$

Table 5 shows the efficiency of the algorithms in the verification of individuals. On the one hand, the semi-automatic algorithm has an average precision of 86.53% for the bird species with noncomplex vocalizations and a maximum of about 55% from species with complex vocalizations. The automatic algorithm has an average precision of 81.93% for the bird species with noncomplex vocalizations and a maximum of about 85.71% from species with complex vocalizations.

**Table 5.** Results of individuals identification performance test (%). Species with complex vocalizations are written in bold.

| Species | Semi-Automatic Individual Verification Algorithm | Automatic Individual Verification Algorithm |
|---|---|---|
| (a) *Automolus rubiginosus* | 85.58 | 75.84 |
| (b) *Synallaxis erythrothorax* | 81.6 | 76 |
| (c) *Cardinalis cardinalis* | 96.94 | 92.98 |
| (d) **Cercomacra tyrannina** | 25 | 82.81 |
| (e) **Hylorchilus sumichrasti** | 55 | 56.5 |
| (f) *Myiozetetes similis* | 82 | 82.9 |
| (g) **Pitangus sulphuratus** | 25 | 75.78 |
| (h) **Psarocolius montezuma** | 47.6 | 85.71 |
| (i) **Amazona viridigenalis** | 29.4 | 55.55 |

## 5. Discussion

### 5.1. Identification and Results Analysis

Table 2 shows the probability of correct rejection of the algorithms. Since there are many possible comparatives between bird species, the main objective of this table is to show the algorithm's capability to reject individuals from impostor species. Note that the described performance (over 80% in almost all the identification comparatives) correspond to the automatic algorithm. The semi-automatic performance varies depending on the capability of the user to determine different species by analyzing the spectrogram.

Table 3 shows the evaluation of the semi-automatic algorithm, where the number of comparatives correspond to the $TA + FA + TR + FR$ matchings. The variability of comparatives is due to the variability of audio data available for each species (see Table 1). Since it is a combinational process, the number of comparatives carried out using the semi-automatic algorithm was only a random sampling. On the other hand, Table 4 shows the same information as in Table 3 but using the automatic algorithm. In this case, the number of comparatives was the total of the combinational process.

The research hypothesis states that the BLPOC function can be used to identify individuals. Table 5 shows the precision results (based on previous tables). Notice that the results differ depending on the bird species and that, although the average stated is 86.53%, the results are given by specifying the individual species performance. The difference between both algorithms' precisions is minimum for noncomplex vocalizations. On the other hand, the automatic algorithm shows a better precision (in comparison with the semi-automatic algorithm) for complex vocalizations. A possible explanation that supports such results is based on the observer identification of the information in the spectrogram. Since this is a subjective selection, the user may omit important information by unselecting this information in the spectrogram (see Figure 5). However, a trained and experienced observer may distinguish the primal spectrogram information.

Thus, the obtained results represent a proof-of-concept and investigating more complex acoustic scenarios is needed to obtain the reliability of the algorithm.

### 5.2. Bird Individual Identification and the Complexity Classification

Even though the automatic algorithm shows better efficiencies in complex vocalizations (see Table 5), this is mainly due to the algorithm's discarding ability for impostor matching. In this sense, a more specialized automatic region extraction algorithm may improve the efficiency of the proposed

automatic algorithm for genuine matching. As expected, the semi-automatic algorithm shows better efficiencies in non-complex vocalizations because the main information from the bird vocalization in the recordings is easy to extract from the spectrogram.

According to the efficiency of the algorithm and the spectrogram characteristics of each vocalization, some differences about what is (or is not) a complex sound (for example, variations of time–frequency and number and type of syllables) could be established. This classification term does not exist in the literature, but it is presented in this article to make a fairer comparison of the algorithm (and for possible bird identifications that could be carried out in future investigations).

### 5.3. Recording Conditions Consideration and Bird Species in the Database

Due to the different sound qualities of the recordings, the collected records do not have a specific recording condition (even among records of individuals of the same bird species). In this context, the proposed algorithm can verify an individual regardless of the difference in the recording conditions (a common condition in field recordings).

There was not biological consideration about the selection of the bird species in the database. This lack of consideration may introduce biases; therefore, in future work, it could be interesting to adopt some more objective criteria at the selection of bird species.

### 5.4. Application Areas

It is still not well understood if the capture of an individual is dangerous to the habitat; in any case, the health of the birds is paramount. Therefore, non-invasive, passive methods are to be preferred (to avoid the common problems of capture such as the stress or death of the individuals) [5,9]. On the other hand, as mentioned in the Introduction, obtaining a large amount (and high quality) of vocalizations from an individual to apply the statistical methods efficiently is nowadays a complex challenge. In this context, the proposed algorithm can provide an identification evaluation by only analyzing two samples from individuals (a limited data condition), implying a time-cost reduction and a new easy way to monitor biodiversity. Although the proposed method does not extract the same features as those from a captured individual, it is a non-invasive method for individual identification that, in addition to the application in counting, could be used to determine the dispersion of the habitats and the mapping of the territories.

Due to the particular skills of experienced observers, it is recommendable that only these qualified observers participate in biodiversity monitoring programs (to guarantee reliability on the data) [5]. Although this is the desired condition, there is often a lack of available qualified observers mainly because of the amount of time consumed by the identifications [7]. On the other hand, training new observers requires from two to three weeks in an intensive program (depending on the number of species to be identified) [9]. The acoustic identification of birds has shown greater efficiency than visual identification, especially because acoustic identification maximizes the visual observations [7]. For that reason, to increase the efficiency of identifications, it is recommended to combine visual and acoustic identification techniques. In this context, the proposed semi-automatic identification algorithm could be used as a valuable tool that can be applied together with traditional counting techniques. Since it is based on the identification of patterns in the spectrogram, the methodology could be used for the training of new observers and to help experienced observers with the reliability of avian counts through the search methodologies.

## 6. Conclusions

Based on the BLPOC function as an efficient method traditionally used in human identification by image matching, in this paper, a new technique for the acoustic individual identification in birds is proposed. Although many methods can provide bird species identification, the proposed method focuses on the individual identification. The results support the proposed algorithm as a method for the acoustic individual identification that can provide an identification evaluation by only analyzing

samples from individuals. Two different methods for the identification are presented, depending on the structure of the vocalization extraction. From the results, it can be concluded that the efficiencies of the algorithm depend on the complexity of the vocalizations. Although complexity of vocalization is the only term used in this paper, it could represent a fair comparison method for bioacoustic algorithms due to the variability of sounds produced by birds (and future work may determine a better metric to determine which sounds are complex and which are not complex). In addition to the individual identification, the acoustic identification technique proposed could be used together with traditional counting techniques to determine the dispersion of the habitats and the mapping of the territories.

**Author Contributions:** Conceptualization, J.V.; Formal analysis, R.R.; Investigation, A.D.P.; Methodology, E.G.; Resources, D.A.; Software, A.B. and G.M.; and Supervision, J.I.D.l.R. All authors have read and agreed to the published version of the manuscript.

**Funding:** This research was funded by PROFEXCE 2020, Secretaria de Educación Pública del Gobierno de México, and PNPC-CONACYT.

**Acknowledgments:** The authors would like to thank CONACYT for the scholarship received by Angel D. Pedroza; the Programa de Doctorado en Ciencias de la Ingeniería from the Universidad Autónoma de Zacatecas; and finally the people who recorded bird vocalizations in Xeno-canto and the bird sound library of Mexico from INECOL.

**Conflicts of Interest:** The authors declare no competing interests.

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
