# Peer review of "Acoustic Individual Identification in Birds Based on the Band-Limited Phase-Only Correlation Function"

_applsci, doi:10.3390/app10072382_

Round 1

Reviewer 1 Report

The authors propose an algorithm to distinguish individuals of bird species to ease the labor-intensive task of monitoring abundance, diversity, migration and dispersion of birds for ecosystem health assessments. The stated motivation is clear and has a well-researched foundation accompanied with relevant references. The overall writing style is good, but requires some revisions especially for commas (there are too many and often misplaced) as well as for some hard to read sentence like: “the variability among bird species and its songs (and calls), makes difficult the automatic extraction of the bird vocalization from the records” or “Since this evaluation criterion represents a combinatorial process between records, only was used a partial selection of the records for the test of the semi-automatic algorithm”

The authors base their idea on the known BLPOC technique often used for fingerprinting in various scenarios. The general idea of this algorithm is described in detail and very thorough verbalizations of the algorithm with all relevant settings and variables make it somewhat easy to follow and understand. The overall structure of the paper is good but sometimes lacks some clarity. E.g. the differences in figure 1 are only minor between the two sides – this could be one single figure to better highlight the difference. Also, Figure 10 seems to be a bit out of place. The authors could maybe cut down on figures or put more of them on one page? Additionally, table 2 is referenced on page 11 and shown on page 13 with no further explanation which requires scrolling or flipping pages.

I also noticed some minor mistakes (?), e.g. Xeno-canto has more than 500K recordings, not just 192K and the Dusky Antbird is Cercomacroides tyrannina according to World List (which taxonomy was used to reference bird species?).

 This leads me to one of the major concerns that I have with this paper: How did the authors come up with the selection of birds for the evaluation? Was there any biological or ecological consideration of which birds might occur at the same time and location? Would it be better to evaluate with species that often occur (and even vocalize) at the same time? I would also like to see species with similar vocalizations and those with extremely complex structures (or sequences) like the Brown Thrasher or Northern Mockingbird.

Additionally, even very “simple” bird species have at least one song and one call – how do you correlate two vocalizations to one individual if using fingerprints? Is that even possible, or feasible? It also appears that the (manual, semi-supervised) selection process of signal sequences (bird vocalizations) seems to be labor intensive – which is opposite of what the authors claimed.

I also have to admit that I do not understand the evaluation mode shown in Table 2. An algorithm that simply rejects all vocalizations would score a perfect 100% if only tested for rejecting “not generated from a specific bird species”. From my view, the better way of evaluating would have been the acceptance rate, not rejection. I had a really hard time to grasp the mode and results of the evaluation in section 4 and I would strongly advise the authors to re-structure this section and to consider a different mode (or maybe just presentation) of the evaluation. I am unsure if the authors confuse species and individual recognition throughout the text. The authors claim “that the proposed algorithm can provide an identification evaluation only by analyzing two samples from individuals” – but I fail to see that. I cannot comprehend the evaluation. Fingerprinting for species classification would not make any sense due to the high (and ever changing) complexity of bird vocalizations. However, it would make sense to recognize individuals to estimate density within one species. But how do the authors create a collection of different individuals of one species with Xeno-canto recordings? There is no way of knowing how many individuals of one species are represented in the dataset. It is nice to see some positive and negative examples of the matching algorithm but it would be better to at least see the corresponding spectrograms. The way the results are presented in this paper is not satisfactory. It is incredibly hard to judge the systems performance.

I am in line with the claim that fingerprinting could help to distinguish individuals of one species. But the presented dataset and evaluation mode do not support the claim that the presented algorithm excels at that task. For species classification, other algorithms cope with hundreds, even thousands of species – none of them ever tackles the task of individual recognition. This paper could be a valuable addition to the field of bioacoustics if the authors can show that their technique actually works. I would advise the authors to re-structure the evaluation process and to provide easly understandable evaluation measures that support the claims that were made. I am looking forward to reading a revised version of this (very promising) paper.

Author Response

Paper: "Acoustic individual identification in birds based on the band-limited phase-only correlation function"

We want to thank all reviewers for their constructive feedback and for reviewing our work. In the sequel, we provide answers to each comment. Furthermore, we made the modifications in the manuscript.

Some of the comments lead us to know that our first version of the document was incomplete. We think that this final version is better than the previous one thanks to the comments made by reviewers.

Reviewer 1

1.-The overall writing style is good, but requires some revisions especially for commas (there are too many and often misplaced) as well as for some hard to read sentence like: “the variability among bird species and its songs (and calls), makes difficult the automatic extraction of the bird vocalization from the records” or “Since this evaluation criterion represents a combinatorial process between records, only was used a partial selection of the records for the test of the semi-automatic algorithm”

Action: Corrected in the paper

2.- The overall structure of the paper is good but sometimes lacks some clarity. E.g. the differences in figure 1 are only minor between the two sides – this could be one single figure to better highlight the difference.

Action: Figure 1 corrected in the paper

3.- Also, Figure 10 seems to be a bit out of place. The authors could maybe cut down on figures or put more of them on one page? Additionally, table 2 is referenced on page 11 and shown on page 13 with no further explanation which requires scrolling or flipping pages.(reedit figures//clarify explanation of the figure).

Action: Figure modificated

4.- I also noticed some minor mistakes (?), e.g. Xeno-canto has more than 500K recordings, not just 192K:

Action: Corrected in the paper

5.- and the Dusky Antbird is Cercomacroides tyrannina according to World List (which taxonomy was used to reference bird species?).

Action: Modified table and included reference [31]

6.-This leads me to one of the major concerns that I have with this paper: How did the authors come up with the selection of birds for the evaluation?:

Response: There is not a special consideration about the selection of bird species. Given that there is a wide variety of songs available in databases, we quickly searched some audios that were easy to identify acoustically, which were different from each other to test the system and that there was a description by the person who recorded it.

7.- Was there any biological or ecological consideration of which birds might occur at the same time and location?

Response: The only consideration was that the recording had acoustically easily identifiable vocalizations and had no other vocalizations from other species. We consider the descriptions of each audio file in the database.

8.- Would it be better to evaluate with species that often occur (and even vocalize) at the same time? .”

Response: In this case, the algorithm will be ineffective since it is necessary to delimit where a vocalization is located. The location (and extraction) of vocalizations is of paramount importance for the efficiency of the algorithm.

9.- I would also like to see species with similar vocalizations and those with extremely complex structures (or sequences) like the Brown Thrasher or Northern Mockingbird

Response: For future work, it would be very interesting to make a comparison such as those mentioned

10.- Additionally, even very “simple” bird species have at least one song and one call – how do you correlate two vocalizations to one individual if using fingerprints? Is that even possible, or feasible?

Response: BLPOC function was recently used in speaker verification. In this article is applied to individual identification.//Action: Article cited in the document//section 3 modificated.

11.- It also appears that the (manual, semi-supervised) selection process of signal sequences (bird vocalizations) seems to be labor intensive – which is opposite of what the authors claimed. (explain the main reason for this situation)

Response: The localization of the vocalizations is a difficult task to perform automatically for the algorithm. Since the task of automatic identification is outside the limits of the research presented, it was decided to develop an algorithm that was intuitive and also represents an analysis tool for the researchers. As mentioned in the introduction many of the total automation tasks in this area have low efficiency levels. Therefore, this algorithm aims to provide a tool for human supervision to carry out the identification

12.- I also have to admit that I do not understand the evaluation mode shown in Table 2. An algorithm that simply rejects all vocalizations would score a perfect 100% if only tested for rejecting “not generated from a specific bird species”. From my view, the better way of evaluating would have been the acceptance rate, not rejection:

Response: Since the function of the algorithm is to verify individuals, the first complexity scale is to verify that the algorithm does not confuse individuals of another species with which it is being tested. This is how specimens were compared against a specific one and it was verified if the algorithm was able to reject individuals that did not belong to the same species//Action:Section 3 and 4 modified.

13.-I had a really hard time to grasp the mode and results of the evaluation in section 4 and I would strongly advise the authors to re-structure this section and to consider a different mode (or maybe just presentation) of the evaluation. 19.-The way the results are presented in this paper is not satisfactory. It is incredibly hard to judge the systems performance (simplify the results in the paper). 20.-I am in line with the claim that fingerprinting could help to distinguish individuals of one species. But the presented dataset and evaluation mode do not support the claim that the presented algorithm excels at that task. 21-This paper could be a valuable addition to the field of bioacoustics if the authors can show that their technique actually works. 22.-I would advise the authors to re-structure the evaluation process and to provide easly understandable evaluation measures that support the claims that were made.

Action: Section 3 and 4 modified//Response: Please, check modified sections

14.-I am unsure if the authors confuse species and individual recognition throughout the text.(check on the hole author).

Action: Verified/corrected in the new version of the article

15.-The authors claim “that the proposed algorithm can provide an identification evaluation only by analyzing two samples from individuals” – but I fail to see that. I cannot comprehend the evaluation

Action: Section 3 modified/Response: Please, check section 3

16.-Fingerprinting for species classification would not make any sense due to the high (and ever changing) complexity of bird vocalizations. However, it would make sense to recognize individuals to estimate density within one species

Action: Corrected in the article

17.-But how do the authors create a collection of different individuals of one species with Xeno-canto recordings? There is no way of knowing how many individuals of one species are represented in the dataset

Response: The database collected was obtained through inspection. This limitation on the amount of audios and the detailed description of them is a very important issue for this type of research. If there were well-identified audios of different individuals, it would be much easier to determine the behavior of the algorithm in question. This was clarified in the document after this comment.//Action: Section modified (clarified)

18.-It is nice to see some positive and negative examples of the matching algorithm but it would be better to at least see the corresponding spectrograms.

Response: this information is pertinent but we want to show the behavior of the BLPOC function. Some types of vocalizations to consider are included in Figure 14.

Reviewer 2 Report

I would like to thank the authors for this highly interested paper on the use of BLPOC for identifying species and individuals from their calls. I believe it will be valuable to bird-researchers and other bioacousticians with an interest in passive acoustic monitoring methods. This is a genuinely exciting potential method for passive acoustic monitoring.

However, it is currently impossible to assess how well the algorithm performed the task as there are no details on the number of individuals or calls included and how successful the algorithm was at extracting calls from the recordings. This really needs to be remedied so that I can properly assess how well the algorithms worked and the potential for the future. I understand the focus was on the acoustic analysis, which is very well detailed, but the statistics are not sufficiently described for proper assessment and material appears in the results that was not presented in the methods.

Also, the writing style, English, and general presentation need a lot of revision as currently it is difficult to follow and there are issues with the presentation of the methods and data sources. There are many mistakes in the English and it is quite difficult to read as it stands.

Edits:

The errors in English start in abstract: “that, has been” shouldn’t have a comma, nor should one follow “discarding”. T

Line 6-7 – Please can you reword “The evaluation of the automatic algorithm 7 discarding, shows a performance over the 90% for almost all the identification comparatives”? I can’t quite follow it.

Section 3

The introduction that is presented here really belongs higher up in the main introduction section.

Figure 1 is excellent and very clear. I would suggest one small adjustment – highlight the difference in the second step to make it clear where the two diverge.

Figure 5 is somewhat squashed and difficult to read properly. I would suggest revising it to make it easier to read and show a smaller section of the overall frequency range to make it clearer that you are focusing on the first ~7,000 Hz. In particular I’d make it clearer that the y-axis is x104 Hz as currently this is not obvious. The legend should also reflect the settings used and allow the reader to recreate the spectrogram.

Section 4 – Results

The source of the acoustic records should appear in the methods, not here, and the number of calls per individual should also be presented.

You also need to make it much clearer how you knew that the calls were from the same individual.

The complex and non-complex categorisation needs to be introduced earlier. What counts as which? And how do you determine if a bird has a complex vocalisation?

Table 1 is helpful, but incomplete. Does each record represent a different individual? And how many calls are there present on each record? It’s impossible to assess how well the algorithm performed if we don’t know what it was run on.

Section 5.3

How did you establish the same bird was present on different recordings? This information needs to appear earlier than the discussion.

How easy is it to record the birds under the same conditions? Do you mean on to the same type of recorder with the same set-up or the distance between bird and recorder? If the latter, that’s much harder to achieve.

Section 5.4

The first two sentences would be better reversed. Mist netting has a negative effect on the caught bird’s welfare, therefore non-invasive, passive methods are to be preferred.

Line 211 – If anything, nowadays it is easier than ever. It has always been a complex challenge.

If you are going to discuss survey methods in the discussion, I would advise that you add a section on them to the introduction.

Section 6 Conclusion

This is better, showing you fully understand the potential implications of your findings and laying out the advantages.

References

Please check that your references are consistent. Sometimes they are italicised and sometimes not, and quote marks are used erratically.

I strongly feel that this manuscript represents a valuable addition to bird research, and from what I've read, I believe that it suffers only from a lack of description, not rigour.

Author Response

Paper: "Acoustic individual identification in birds based on the band-limited phase-only correlation function"

We want to thank all reviewers for their constructive feedback and for reviewing our work. In the sequel, we provide answers to each comment. Furthermore, we made the modifications in the manuscript.

Some of the comments lead us to know that our first version of the document was incomplete. We think that this final version is better than the previous one thanks to the comments made by reviewers.

Reviewer 2

23.-However, it is currently impossible to assess how well the algorithm performed the task as there are no details on the number of individuals or calls included

Action:Corrected in the paper.

Response:Please, check table

24.- and how successful the algorithm was at extracting calls from the recordings (clarify on the paper if necessary or justify).” 25.-This really needs to be remedied so that I can properly assess how well the algorithms worked and the potential for the future:

Action: Section 3 modified

25.-I understand the focus was on the acoustic analysis, which is very well detailed but the statistics are not sufficiently described for proper assessment.

Action: Section 3 and 4 modified//Response: Please, check sections modified

26.- material appears in the results that was not presented in the methods

Action: Section corrected

27.- The errors in English start in abstract: “that, has been” shouldn’t have a comma, nor should one follow “discarding”. T

Action: Section corrected

28.- Line 6-7 – Please can you reword “The evaluation of the automatic algorithm 7 discarding, shows a performance over the 90% for almost all the identification comparatives”? I can’t quite follow it.

Action:Changed

29.- The introduction that is presented here really belongs higher up in the main introduction section

Action: Paragraph modified

30.- Figure 1 is excellent and very clear. I would suggest one small adjustment – highlight the difference in the second step to make it clear where the two diverge.

Action: Figure Modified

31.- Figure 5 is somewhat squashed and difficult to read properly. I would suggest revising it to make it easier to read and show a smaller section of the overall frequency range to make it clearer that you are focusing on the first ~7,000 Hz. In particular I’d make it clearer that the y-axis is x104 Hz as currently this is not obvious. The legend should also reflect the settings used and allow the reader to recreate the spectrogram

Action:Figures modified

32.-The source of the acoustic records should appear in the methods, not here, and the number of calls per individual should also be presented.

Action:Corrected in the paper.

33.- You also need to make it much clearer how you knew that the calls were from the same individual. (clarify for the xeno canto description). 33.-Table 1 is helpful, but incomplete. Does each record represent a different individual? And how many calls are there present on each record? It’s impossible to assess how well the algorithm performed if we don’t know what it was run on.

Action:Corrected in the paper//Response:Please check section 3 and 4

34.- The complex and non-complex categorisation needs to be introduced earlier. What counts as which? And how do you determine if a bird has a complex vocalization?

Response: The complex category is based on the performance of the algorithm and the spectrogram of each vocalization. As such, this classification does not exist in the literature but this term is introduced in this article in order to make a more fair comparison for the algorithm. Its support  it that it is not possible to obtain a general classification of the algorithm for all species that is being recognized.

Action: Section modified

35.-How did you establish the same bird was present on different recordings? This information needs to appear earlier than the discussion.

Action:Section 3 and 4 modified//Response:Please check the sections

36.- How easy is it to record the birds under the same conditions? Do you mean on to the same type of recorder with the same set-up or the distance between bird and recorder? If the latter, that’s much harder to achieve.

Action:Clarified in the document//Response: that’s  right.

37.- The first two sentences would be better reversed. Mist netting has a negative effect on the caught bird’s welfare, therefore non-invasive, passive methods are to be preferred.

Action:Paragraph modified

38.- If anything, nowadays it is easier than ever. It has always been a complex challenge:

Action:Paragraph modified

39.- If you are going to discuss survey methods in the discussion, I would advise that you add a section on them to the introduction.

Action: Section reviewed

40.- This is better, showing you fully understand the potential implications of your findings and laying out the advantages.

Action:Section modified

41.- Please check that your references are consistent. Sometimes they are italicised and sometimes not, and quote marks are used erratically. (change)(review methrics and form of the paper):

Action:Section verified

Reviewer 3 Report

In general, I consider the approach interesting and promising. But as the authors at least suggest the method is not suitable for birds with higher intra-individual variability. Their statement that even different species can be discriminated sounds almost ironic.

How is the reader going to profit from the paper?
Why do you not provide coding details (e.g. R script based on warbleR)?

There are many issues where English grammar is incorrectly or at least inappropriately used. The MS needs to undergo thorough language editing.

Few minor issues:
153: better: citizen-science database (all individual recordings used need to be listed in a supplementary table)
159-163 and Species columns of Tables 1 and 3: italicize all scientific bird names (lower case after first letter of genus name!), thus use another means to highlight species in Table 3
244-245 not suitable to show who did what, especially the actual technical work and programming

Author Response

Paper: "Acoustic individual identification in birds based on the band-limited phase-only correlation function"

We want to thank all reviewers for their constructive feedback and for reviewing our work. In the sequel, we provide answers to each comment. Furthermore, we made the modifications in the manuscript.

Some of the comments lead us to know that our first version of the document was incomplete. We think that this final version is better than the previous one thanks to the comments made by reviewers.

Reviewer 3

43.- Their statement that even different species can be discriminated sounds almost ironic (erase from the paper since discarding statistics are not enough for establish this task).

Action: Paragraph modified//Response: that’s  right

44.- How is the reader going to profit from the paper?

Response: Although there are a lot of methods that can provide bird species identification, the proposed method focuses on the individual identification. Results support the proposed algorithm as a method for the acoustic individual identification that can provide an identification evaluation only by analyzing samples from individuals.

45.- Why do you not provide coding details (e.g. R script based on warbleR)?

Response: Although it is interesting, it exceeds the limits of the document. This could be done for another article.

46.- : better: citizen-science database

Action: Table 1 modified

47.- and Species columns of Tables 1 and 3: italicize all scientific bird names (lower case after first letter of genus name!), thus use another means to highlight species in Table 3

Action:Table 3 modified

48.- not suitable to show who did what, especially the actual technical work and programming

Action: Paragraph erased

Round 2

Reviewer 1 Report

Thanks for providing the updated articles and clarifying comments – I really appreciate your work. While reading the article for the second time, some things still confused me:

  • Commas were removed but the English got worse (even in the abstract; e.g. “The proposed technique has two variants: one automatic and other semi-automatic.”, “First, it is necessary a vocalization database of a few individuals (to evaluate the proposed methodology)”
  • If all birds in the dataset are passerines, you could get rid of this one column to save space
  • Still, information on how you selected species for your dataset is missing, you mentioned in the comments that you decided to only use species that are easy enough to identify, which is a huge bias and should be mentioned in the text and discussion
  • In the current state (and that’s also something you mentioned in the comments), this paper should be considered a proof-of-concept and not a full-sized assessment of BLPOC for bird identification – from my perspective, this is not always clearly stated in the text and might confuse readers who expect a more thorough investigation
  • I see that some figures have been rearranged, which is good, the review format with line numbers still messes up the layout but that’s not your fault
  • There are a number of attempts to measure and/or describe the complexity of bird song (e.g. by counting syllable types). That should be mentioned in the paper. Yet, admittedly, all methods seem to have a certain bias and are not necessarily objective
  • It is clear to me now, how and why the first step of the evaluation (Table 2) was conducted. However, why didn’t you test the TRR for each species with itself? If your proposed Algorithm rejects recordings although it is the same species, that should be mentioned in the article. Again, in my view, an algorithm that simply rejects all impostors, would score 100% TRR using your evaluation mode? If not, the article should explain why – again, I have difficulties to fully understand the evaluation despite your efforts to make it clearer.
  • Additionally, it would be nice to explore some problematic cases like f+c with only 69% - why does this happen, what is your hypothesis?
  • The most important presentation of results seems to be Table 3 because it eventually assesses the performance for individual recognition – I would really hope to see more details (why not a similar Table as in Table 2?) and a more detailed discussion. Yet, it is clearer to me know how both evaluations were intended, which is a step in the right direction

Again, individual recognition could be an important method in bioacoustics and if you can proof that fingerprinting might work, it would be a great advancement of the filed. For now, you need to state that your article is considered a proof-of-concept and that you plan to investigate more complex acoustic scenarios (soundscapes?) in the future.

Author Response

Paper: "Acoustic individual identification in birds based on the band-limited phase-only correlation function"

Thank all reviewers for their constructive feedback and for reviewing our work. In the sequel, we provide answers to each comment. Furthermore, we made the modifications in the manuscript.

Comments lead us to know that our second version of the document was incomplete. This final version is better than the previous one thanks to the comments made by reviewers.

About the use of “Track Changes function” it is still confusing how to implement this function in latex document from Word. We hope editors can identify changes made at the original document.

Reviewer 1

1.-Commas were removed but the English got worse (even in the abstract; e.g. “The proposed technique has two variants: one automatic and other semi-automatic.”, “First, it is necessary a vocalization database of a few individuals (to evaluate the proposed methodology)”.

Action: revised and republished full text (line 1-342)

2.-If all birds in the dataset are passerines, you could get rid of this one column to save space.-

Response: This modification appears to be important and was made in the document.

Action: Table 1 modified in the text (page 3).

3.-Still, information on how you selected species for your dataset is missing, you mentioned in the comments that you decided to only use species that are easy enough to identify, which is a huge bias and should be mentioned in the text and discussion.-

Response: As indicated, this consideration may introduce bias; this is clarified in the text. There weren't any biological consideration about the selection of the bird species in the database. This lack of consideration may introduce biases, therefore in future work, it could be interesting to adopt some more objective criteria at the selection of bird species.

Action: Clarified in lines  227-229.

4.-In the current state (and that’s also something you mentioned in the comments), this paper should be considered a proof-of-concept and not a full-sized assessment of BLPOC for bird identification – from my perspective, this is not always clearly stated in the text and might confuse readers who expect a more thorough investigation:

Response: This comment leads us to restructure the limits of the research.

Action: Clarified in line 10-13, 47-50, 208-209,

5.-There are a number of attempts to measure and/or describe the complexity of bird song (e.g. by counting syllable types). That should be mentioned in the paper. Yet, admittedly, all methods seem to have a certain bias and are not necessarily objective.

Response: According to the efficiency of the algorithm and the spectrogram characteristics of each vocalization, it could establish some differences about what is (or is not) a complex sound (for example, variations of time-frequency, number and type of syllables). This classification term does not exist in the literature, but it is presented in this article to make a fairer comparison of the algorithm (and for possible bird identifications that could be carried out in future investigations)

Action: Clarified in line 217-221

6.-It is clear to me now, how and why the first step of the evaluation (Table 2) was conducted. However, why didn’t you test the TRR for each species with itself? If your proposed Algorithm rejects recordings although it is the same species, that should be mentioned in the article. Again, in my view, an algorithm that simply rejects all impostors, would score 100% TRR using your evaluation mode? If not, the article should explain why – again, I have difficulties to fully understand the evaluation despite your efforts to make it clearer.

Response: This question guides us to create new tables and a deeper description of the algorithm, thanks for the comment.

Action: New Tables: Table 3 and Table 4 (page 13-14)

            New equation: Equation 16 (page 14)

            Line modification: 159-183

7.-Additionally, it would be nice to explore some problematic cases like f+c with only 69% - why does this happen, what is your hypothesis

Response: The metrics change to explore the possible explanation

Action: New Tables: Table 3 and Table 4 (page 13-14)

            New equation: Equation 16 (page 14)

            Line modification: 198-209

8.-The most important presentation of results seems to be Table 3 because it eventually assesses the performance for individual recognition – I would really hope to see more details (why not a similar Table as in Table 2?) and a more detailed discussion. Yet, it is clearer to me know how both evaluations were intended, which is a step in the right direction.

Response: The metrics change to explore the possible explanation

Action: New Tables: Table 3 and Table 4 (page 13-14)

            New equation: Equation 16 (page 14)

            New section “5.1. Identification and results analysis” (line 186-209).

Reviewer 2 Report

I would like to thank the authors for their revisions in the light of my comments. The manuscript is much improved and I appreciate all of their work on it.

Minor points:

The language is generally much improved but still remains somewhat difficult to read in places, e.g. line 34 “Some of the statistical techniques for automatic speaker recognition commonly are also applied to individual identification of animals” could be better phrased as “Some of the statistical techniques COMMONLY USED for speaker recognition CAN also be applied to THE individual identification of animals.” Ditto line 41 where either “we propose” or “a new technique is proposed” would be better English. This is not the only example of where the language needs to be altered, but it was one of the first I found.

Lines 50-52 show results in the introduction and should be removed.

Line 70 – This sentence is missing its verb.

Table 1 is much improved, thank you.

Figure 5 is also much better, well done.

Figure 14 is excellent and answers my concern about complex vs non-complex perfectly.

Author Response

Paper: "Acoustic individual identification in birds based on the band-limited phase-only correlation function"

Thank all reviewers for their constructive feedback and for reviewing our work. In the sequel, we provide answers to each comment. Furthermore, we made the modifications in the manuscript.

Comments lead us to know that our second version of the document was incomplete. This final version is better than the previous one thanks to the comments made by reviewers.

About the use of “Track Changes function” it is still confusing how to implement this function in latex document from Word. We hope editors can identify changes made at the original document.

Reviewer 2

9.-The language is generally much improved but still remains somewhat difficult to read in places, e.g. line 34 “Some of the statistical techniques for automatic speaker recognition commonly are also applied to individual identification of animals” could be better phrased as “Some of the statistical techniques COMMONLY USED for speaker recognition CAN also be applied to THE individual identification of animals.” Ditto line 41 where either “we propose” or “a new technique is proposed” would be better English. This is not the only example of where the language needs to be altered, but it was one of the first I found

Action: revised and republished full text (line 1-342)

10.-Lines 50-52 show results in the introduction and should be removed:

Action: Lines removed

11.-Line 70 – This sentence is missing its verb..-

Action: sentence completed:”to collect”.